# Systematic Evaluation of Attribution Methods: Eliminating Threshold Bias and Revealing Method-Dependent Performance Patterns

## Abstract

Attribution methods explain neural network predictions by identifying influential input features, but their evaluation suffers from threshold selection bias that can reverse method rankings and undermine conclusions. Current protocols binarize attribution maps at single thresholds, where threshold choice alone can alter rankings by over 200 percentage points. We address this flaw with a threshold-free framework that computes Area Under the Curve for Intersection over Union (AUC-IoU), capturing attribution quality across the full threshold spectrum. Evaluating seven attribution methods on dermatological imaging, we show single-threshold metrics yield contradictory results, while threshold-free evaluation provides reliable differentiation. XRAI achieves 31% improvement over LIME and 204% over vanilla Integrated Gradients, with size-stratified analysis revealing performance variations up to 269% across lesion scales. These findings establish methodological standards that eliminate evaluation artifacts and enable evidence-based method selection. The threshold-free framework provides both theoretical insight into attribution behavior and practical guidance for robust comparison in medical imaging and beyond.

## 1 Introduction

Attribution methods have been developed to explain neural network predictions by identifying which input features most influence model outputs. However, their evaluation suffers from a fundamental methodological flaw: arbitrary threshold selection bias that can reverse performance rankings and undermine scientific conclusions. Current evaluation protocols rely on single threshold binarization of continuous attribution maps, where threshold choice alone can alter method rankings by over 200 percentage points, making comparative studies unreliable. The threshold selection problem emerges from the diversity of attribution approaches and their distinct response characteristics. Gradient-based methods like Integrated Gradients produce concentrated, high-magnitude attributions that are optimally evaluated at low thresholds, while perturbation-based approaches like LIME generate more diffuse attributions favoring higher thresholds (Ribeiro et al., 2016; Sundararajan et al., 2017). Consequently, threshold choice predetermines evaluation outcomes independently of actual attribution quality, introducing systematic bias that compromises method comparison reliability. Recent work has exposed critical evaluation failures across explainable AI research. Input invariance violations have been demonstrated where methods produce different explanations for identical model outputs (Kindermans et al., 2017). Sanity checks reveal that some widely used techniques are independent of model parameters and training data (Adebayo et al., 2018). Contradictory results between popular evaluation metrics further highlight fundamental assessment limitations (Nielsen et al., 2023). These findings indicate that current evaluation practices may reflect measurement artifacts rather than genuine method performance differences. This work addresses threshold selection bias through a comprehensive evaluation framework that eliminates arbitrary threshold choice. A threshold-free assessment protocol using Area Under the Curve metrics for Intersection over Union (AUC-IoU) is introduced, computing attribution quality across the complete threshold spectrum rather than at single arbitrary points. The framework is validated through systematic evaluation of seven attribution methods representing major paradigms: gradient-based (Integrated Gradients variants), activation-based (Grad-CAM), perturbation-based (LIME), and region-based (XRAI) ap-

proaches. Empirical analysis on dermatological imaging reveals that conventional single-threshold evaluation leads to contradictory method rankings, with performance differences exceeding 235 percentage points depending solely on threshold selection. Statistical validation using Wilcoxon signed-rank tests with multiple comparison correction establishes that threshold-free evaluation enables reliable method differentiation, revealing that XRAI achieves 31% improvement over LIME and 204% improvement over vanilla Integrated Gradients. Size-stratified analysis demonstrates that method performance varies substantially based on lesion characteristics, with improvement factors ranging from 0% to 269% across different scales. These contributions establish methodological standards for attribution evaluation that eliminate evaluation artifacts and enable evidence-based method selection in critical applications. The threshold-free framework provides both theoretical understanding of attribution method behavior and practical guidance for reliable technique comparison across diverse domains.

## 2 RELATED WORK

### 2.1 ATTRIBUTION METHOD PARADIGMS

Four primary paradigms have been established for neural network attribution. Gradient-based approaches compute feature importance through backpropagation, with Integrated Gradients addressing fundamental axiom violations in simple gradient methods by ensuring Completeness and Implementation Invariance through path integration (Sundararajan et al., 2017). Noise reduction techniques like SmoothGrad improve visual quality by averaging attributions across multiple noisy input versions (Smilkov et al., 2017). Activation-based methods use intermediate network representations to generate localization maps. Grad-CAM produces class-discriminative visualizations by combining gradient information with activation maps, providing broad architectural compatibility without requiring structural modifications (Selvaraju et al., 2016). Perturbation-based approaches learn interpretable models locally around specific predictions. LIME explains arbitrary classifiers by fitting linear models to prediction changes under feature perturbations, enabling model-agnostic explanations across diverse domains (Ribeiro et al., 2016). Region-based methods extend pixel-level attributions to semantically coherent segments. XRAI builds upon Integrated Gradients but operates on image regions rather than individual pixels, addressing fragmentation issues through iterative region selection based on attribution density (Kapishnikov et al., 2019).

### 2.2 EVALUATION METHODOLOGY CHALLENGES

Critical limitations in attribution evaluation have been systematically documented. Input invariance failures demonstrate that methods produce different explanations when constant shifts are applied to inputs despite identical model outputs (Kindermans et al., 2017). Model parameter randomization tests reveal that some techniques function independently of learned representations, suggesting they detect input structure rather than model behavior (Adebayo et al., 2018). Comprehensive benchmarking efforts have revealed contradictions between evaluation metrics. The EvalAttAI framework demonstrates that popular Deletion and Insertion metrics yield contradictory results, with methods performing well on one showing poor performance on the other (Nielsen et al., 2023). Medical imaging evaluations consistently show that attribution methods achieve only moderate alignment with expert annotations, with best-performing approaches reaching 41% accuracy in highlighting diagnostically relevant regions (Cerekci et al., 2024). Standardization challenges persist across evaluation practices. Threshold selection bias has been identified in medical image segmentation evaluation, where arbitrary cutoff choices dramatically affect performance interpretation (Müller et al., 2022). However, systematic analysis of threshold bias in attribution evaluation has received limited attention, representing a critical gap in methodological understanding.

## 3 METHODOLOGY

### 3.1 DATASET AND EXPERIMENTAL DESIGN

In this study, the HAM10000 dataset (10,015 dermoscopic images) was used for binary classification (melanoma vs. non-melanoma). Images were resized to 224×224 and standardized using ImageNet statistics; segmentation masks were binarized for evaluation. A stratified 70/15/15 split preserved

class balance, as seen in Table 1. For attribution evaluation, we constructed a 500-image test subset including all melanoma cases (n=167) and 333 randomly sampled non-melanoma cases, ensuring statistical robustness and adequate minority class representation.

Table 1: Dataset distribution across splits.

| Split | Melanoma | Non-Melanoma | Total | Melanoma % | Purpose |
|---|---|---|---|---|---|
| Train | 779 | 6,231 | 7,010 | 11.11% | Model training |
| Validation | 167 | 1,335 | 1,502 | 11.12% | Model validation |
| Test | 167 | 1,336 | 1,503 | 11.11% | Model evaluation |
| Attribution Evaluation | 167 | 333 | 500 | 33.40% | XAI method comparison |

## 3.2 MODEL ARCHITECTURE AND TRAINING

A ResNet-18 pretrained on ImageNet was fine-tuned for binary classification. Early layers were frozen, and layer4 plus the classifier were updated, resulting in 8.4M trainable parameters. Training used Adam (1e-4) with class-weighted cross-entropy to address imbalance, and early stopping (patience=5). Full preprocessing, optimization, and calibration details are provided in Appendix B.

## 3.3 ATTRIBUTION METHOD IMPLEMENTATION

Seven attribution methods representing major explainability paradigms were implemented using the saliency library with consistent preprocessing and model interfaces:

• **Region-based:** XRAI with batch size 20 for computational efficiency.

• **Gradient-based methods:** Four variants of Integrated Gradients were evaluated: (1) Vanilla Integrated Gradients with 25 integration steps, zero baseline, and batch size 20, (2) Blur IG with batch size 20, (3) SmoothGrad IG using GetSmoothedMask with 25 integration steps, zero baseline, and batch size 20, and (4) Guided IG with 25 integration steps, zero baseline, maximum distance 1.0, and fraction 0.5.

• **Activation-based:** GradCAM targeting ResNet-18 layer3[1].conv2 with forward and backward hooks registered for activation and gradient capture during backpropagation.

• **Perturbation-based:** LIME implemented using lime_image.LimeImageExplainer with 1000 perturbations per image, kernel width 1.0, Ridge regression regularization ($\alpha = 10.0$), batch size 32, and random seed 42 for reproducibility.

Temperature-scaled model outputs were utilized for LIME, which relies on probability estimates for perturbation-based explanations. Other attribution methods used the underlying model logits and gradients.

## 3.4 ATTRIBUTION EVALUATION FRAMEWORK

### 3.4.1 THRESHOLD-FREE EVALUATION PROTOCOL

## 3.5 EVALUATION METRICS

Traditional single-threshold evaluation was replaced with comprehensive threshold-free assessment to eliminate arbitrary threshold selection bias. For each attribution map $A$ and ground truth mask $G$, Intersection over Union (IoU) was calculated across 19 uniformly spaced thresholds $\tau \in [0.05, 0.95]$:

$$\text{IoU}(\tau) = \frac{|A_\tau \cap G|}{|A_\tau \cup G|} \tag{1}$$

where $A_\tau$ represents the binarized attribution map at threshold $\tau$ after normalization to $[0, 1]$. IoU calculation included handling of edge cases where the union equals zero, returning a score of 1.0 to avoid division by zero.

The Area Under the IoU Curve (AUC-IoU) was computed using trapezoidal integration:

$$\text{AUC-IoU} = \int_{0.05}^{0.95} \text{IoU}(\tau)\, d\tau \tag{2}$$

### 3.6 THRESHOLD BIAS ANALYSIS

To systematically evaluate threshold selection bias in attribution assessment, AUC-IoU performance was compared against conventional single-threshold evaluation at three representative values: $\tau = 0.3$ (low threshold), $\tau = 0.5$ (medium threshold), and $\tau = 0.7$ (high threshold). These thresholds span the operational range while representing commonly employed evaluation points in existing attribution literature.

For each method-threshold combination, relative performance differences were calculated as:

$$\text{Relative Difference} = \frac{\text{AUC-IoU} - \text{IoU}(\tau)}{\text{IoU}(\tau)} \times 100\% \tag{3}$$

Performance swings were quantified as the absolute difference between extreme threshold evaluations ($\tau = 0.3$ vs $\tau = 0.7$) to measure the full magnitude of evaluation bias introduced by threshold selection.

### 3.7 SIZE STRATIFICATION ANALYSIS

Lesion size was quantified as the number of positive pixels in the original-resolution segmentation masks (768×768) prior to resizing for model input. Size-based stratification employed percentile thresholds: small lesions ($\leq$33rd percentile, $\leq$40,956 pixels), medium lesions (33rd–67th percentile, 40,956–84,880 pixels), and large lesions ($\geq$67th percentile, $\geq$84,880 pixels). This classification enabled analysis of method performance dependencies on lesion scale characteristics, with the evaluation subset containing 133 small, 160 medium, and 207 large lesions.

### 3.8 STATISTICAL ANALYSIS

#### 3.8.1 METHOD PERFORMANCE COMPARISONS

Statistical significance was assessed using Wilcoxon signed-rank tests for pairwise method comparisons, chosen for appropriateness with paired non-parametric data and potential non-normal AUC-IoU distributions. Effect sizes were calculated as median paired differences to properly account for paired observations across the same image set. Multiple comparison correction employed the Holm-Bonferroni procedure controlling family-wise error rate $\alpha = 0.05$ across 21 pairwise tests.

#### 3.8.2 THRESHOLD BIAS STATISTICAL FRAMEWORK

For threshold comparison analysis, paired Wilcoxon signed-rank tests compared AUC-IoU scores against single-threshold IoU scores ($\tau = 0.3, 0.5, 0.7$) across all 500 evaluation images. Holm-Bonferroni correction was applied across 133 statistical comparisons (7 methods × 19 thresholds) to control family-wise error rate. Method ranking stability was assessed by comparing ordinal positions between AUC-based and single-threshold evaluations to identify ranking reversals that could affect clinical deployment decisions.

## 4 RESULTS

### 4.1 MODEL PERFORMANCE AND EVALUATION FRAMEWORK

The ResNet-18 model achieved 91.75% accuracy on the test set, with precision/recall of 0.95/0.96 for non-melanoma and 0.64/0.60 for melanoma cases, establishing sufficient baseline performance for attribution analysis. Attribution methods were evaluated on 500 strategically selected images using threshold-free AUC-IoU scores across 19 threshold levels.

## 4.2 COMPREHENSIVE EVALUATION RESULTS

Comprehensive evaluation revealed substantial performance differences between attribution methods, with XRAI demonstrating clear superiority across all evaluation metrics. Table 2 presents the complete performance ranking with statistical confidence intervals.

Table 2: Attribution Method Performance Summary

| Method | Mean AUC-IoU | Std Dev | 95% CI |
|---|---|---|---|
| XRAI | 0.1844 | 0.1137 | ±0.0100 |
| LIME | 0.1409 | 0.1077 | ±0.0095 |
| SmoothGrad_IG | 0.1174 | 0.0596 | ±0.0052 |
| GradCAM | 0.1146 | 0.0929 | ±0.0082 |
| Blur_IG | 0.0979 | 0.0286 | ±0.0025 |
| Guided_IG | 0.0968 | 0.0379 | ±0.0033 |
| Vanilla_IG | 0.0606 | 0.0411 | ±0.0036 |

XRAI achieved the highest mean AUC-IoU score (0.1844), representing a 31% improvement over LIME (0.1409) and a 204% improvement over Vanilla_IG (0.0606). The performance distribution exhibited clear stratification, with XRAI forming a distinct top tier, followed by LIME and SmoothGrad_IG in the second tier. SmoothGrad_IG demonstrated the lowest performance variability ($\sigma = 0.0596$), indicating consistent attribution quality, while XRAI showed higher variability but maintained superior average performance. Figure 1 demonstrates that the 95% confidence intervals are sufficiently narrow to establish clear performance distinctions between methods. The tight error bars, achieved through evaluation on 500 images, confirm that XRAI's superiority represents genuine performance differences rather than sampling variability.

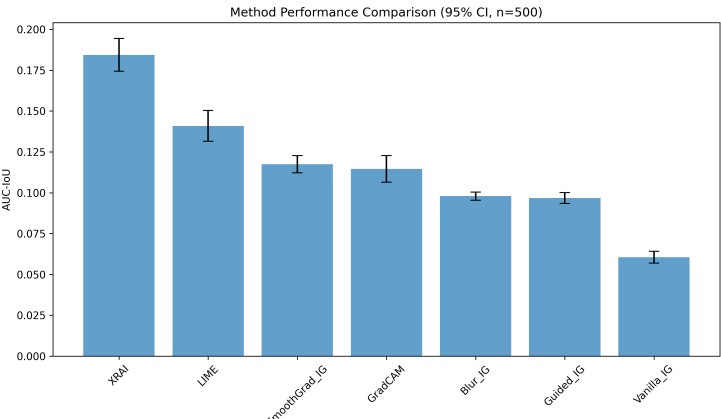

Figure 1: Method performance comparison showing mean AUC-IoU scores with 95% confidence intervals ($n = 500$).

## 4.3 STATISTICAL SIGNIFICANCE ANALYSIS

Statistical testing using Wilcoxon signed-rank tests with Holm-Bonferroni correction revealed that apparent performance differences represent method distinctions rather than random variation. Table 3 summarizes key pairwise comparisons with corrected significance levels.

XRAI significantly outperformed all competing methods ($p < 10^{-17}$), with effect sizes ranging from 0.0391 to 0.1080. The largest effect size occurred in XRAI vs. Vanilla_IG (0.1080), corresponding to the 204% performance difference. Critically, several method pairs showed no significant differences: GradCAM vs. SmoothGrad_IG ($p = 0.156$) and Blur_IG vs. Guided_IG ($p = 0.0747$), indicating that apparent ranking differences may reflect measurement noise rather than genuine distinctions.

Table 3: Statistical Significance of Method Comparisons

| Comparison | p-value | Effect Size | Significance |
|---|---|---|---|
| XRAI vs. LIME | $2.22 \times 10^{-17}$ | 0.0391 | *** |
| XRAI vs. SmoothGrad_IG | $4.14 \times 10^{-38}$ | 0.0443 | *** |
| XRAI vs. GradCAM | $9.36 \times 10^{-27}$ | 0.0631 | *** |
| XRAI vs. Vanilla_IG | $1.51 \times 10^{-83}$ | 0.1080 | *** |
| LIME vs. Vanilla_IG | $2.71 \times 10^{-54}$ | 0.0603 | *** |
| GradCAM vs. SmoothGrad_IG | 0.156 | -0.0129 | ns |
| Blur_IG vs. Guided_IG | 0.0747 | 0.0052 | ns |

## 4.4 Size-Stratified Performance Analysis

Size-stratified analysis revealed method performance dependencies on lesion characteristics that challenge assumptions of uniform method applicability. Table 4 presents performance by lesion size category.

Table 4: Performance by Lesion Size Category

| Method | Small (n=133) | Medium (n=160) | Large (n=207) | Improvement |
|---|---|---|---|---|
| XRAI | $0.106 \pm 0.091$ | $0.160 \pm 0.092$ | $0.254 \pm 0.102$ | 139% |
| GradCAM | $0.046 \pm 0.055$ | $0.099 \pm 0.069$ | $0.171 \pm 0.095$ | 269% |
| LIME | $0.061 \pm 0.069$ | $0.139 \pm 0.109$ | $0.194 \pm 0.095$ | 218% |
| SmoothGrad_IG | $0.083 \pm 0.047$ | $0.106 \pm 0.055$ | $0.149 \pm 0.055$ | 80% |
| Blur_IG | $0.096 \pm 0.036$ | $0.102 \pm 0.030$ | $0.096 \pm 0.020$ | 0% |
| Guided_IG | $0.070 \pm 0.029$ | $0.088 \pm 0.022$ | $0.121 \pm 0.039$ | 72% |
| Vanilla_IG | $0.031 \pm 0.023$ | $0.052 \pm 0.034$ | $0.087 \pm 0.040$ | 183% |

Size-dependent performance variation exceeded expectations, with improvement factors ranging from 0% (Blur_IG) to 269% (GradCAM). XRAI maintained superiority across all size categories with 139% improvement from small to large lesions. GradCAM showed the most dramatic size sensitivity, increasing 269% from worst performance on small lesions (0.046) to competitive performance on large lesions (0.171). For clinically critical small lesions, method selection becomes crucial. XRAI (0.106) substantially outperformed all alternatives, with the performance gap having direct clinical implications where attribution quality impacts diagnostic confidence for challenging small lesion detection.

## 4.5 Threshold-Free Evaluation Insights

Threshold-free evaluation across the complete threshold spectrum ($\tau \in [0.05, 0.95]$) on 500 dermatological images revealed critical limitations of conventional single-threshold approaches that challenge fundamental assumptions underlying current evaluation methodologies.

### 4.5.1 Threshold-Dependent Performance Variability and Ranking Instability

Single-threshold evaluation exhibited extreme sensitivity to threshold selection, with method performance varying dramatically across the evaluation range. Table 5 presents comparative analysis between AUC-IoU and commonly employed single-threshold metrics, revealing systematic evaluation bias.

Table 5: Threshold-Free vs Single-Threshold Performance Comparison

| Method | AUC-IoU | IoU@0.3 | Rel. Diff. | IoU@0.5 | Rel. Diff. | IoU@0.7 | Rel. Diff. |
|---|---|---|---|---|---|---|---|
| XRAI | 0.1844 | 0.2784 | -33.8%*** | 0.2331 | -20.9%*** | 0.1483 | +24.3%*** |
| LIME | 0.1409 | 0.1565 | -10.0%*** | 0.1565 | -10.0%*** | 0.1565 | -10.0%*** |
| SmoothGrad_IG | 0.1172 | 0.1980 | -40.8%*** | 0.1095 | +7.0%*** | 0.0536 | +118.7%*** |
| GradCAM | 0.1146 | 0.1856 | -38.3%*** | 0.1266 | -9.5%*** | 0.0671 | +70.7%*** |
| Blur_IG | 0.0979 | 0.1425 | -31.3%*** | 0.0785 | +24.7%*** | 0.0467 | +109.7%*** |
| Guided_IG | 0.0968 | 0.1508 | -35.8%*** | 0.0788 | +22.8%*** | 0.0412 | +134.8%*** |
| Vanilla_IG | 0.0606 | 0.0904 | -32.9%*** | 0.0422 | +43.5%*** | 0.0200 | +202.7%*** |

*All differences statistically significant: ** $p < 0.001$ (Wilcoxon signed-rank test, Holm-Bonferroni corrected, n=500).*

Individual methods exhibited performance swings exceeding 200 percentage points, with Vanilla_IG showing a 235.6 percentage point variation from $\tau = 0.3$ ($-32.9\%$) to $\tau = 0.7$ ($+202.7\%$). This extreme variability demonstrates that threshold choice alone can determine whether Vanilla_IG appears substantially inferior or superior to threshold-free evaluation.

The systematic bias patterns reveal method-specific evaluation artifacts. Gradient-based methods consistently show negative relative differences at low thresholds, indicating their concentrated attribution patterns are penalized by aggressive binarization. Conversely, positive relative differences at high thresholds suggest these methods benefit from conservative threshold selection. LIME's unique threshold-invariant behavior ($-10.0\%$ across all $\tau$) reflects its superpixel-based approach, making it the only method where single-threshold evaluation provides reliable performance estimation.

### 4.5.2 STATISTICAL VALIDATION AND IMPLICATIONS

All method-threshold combinations showed statistically significant differences ($p < 0.001$) after Holm-Bonferroni correction for 133 multiple comparisons, with corrected p-values ranging from $1.3 \times 10^{-80}$ to $8.9 \times 10^{-5}$. Lower thresholds systematically favor single-threshold metrics (7–41% effect sizes), while higher thresholds favor AUC-based evaluation. These findings demonstrate that traditional single-threshold approaches introduce predictable directional bias and threshold-dependent ranking instabilities that compromise method comparison reliability. The systematic performance variability indicates that previous comparative studies employing single-threshold metrics may have drawn conclusions that are artifacts of threshold selection rather than genuine method performance differences. Threshold-free evaluation protocols are essential for robust attribution method assessment in medical imaging applications, where evaluation reliability directly impacts clinical decision-making confidence.

## 5 DISCUSSION

### 5.1 METHODOLOGICAL IMPLICATIONS AND COMPARISON WITH PRIOR FRAMEWORKS

Threshold selection can reverse method rankings by more than 200 percentage points, exposing a fundamental flaw in current XAI evaluation practices. This bias suggests many published comparisons reflect artifacts of threshold choice rather than true performance differences. Gradient-based methods favor low thresholds, while perturbation-based methods (e.g., LIME) remain threshold-invariant patterns ignored by current single-threshold protocols. Such assumptions of uniform response are invalid, directly undermining meta-analyses and systematic reviews, where differing threshold choices may explain contradictory findings. The threshold-free framework addresses these limitations by evaluating across the complete threshold spectrum. Unlike Deletion and Insertion metrics that produce contradictory results Nielsen et al. (2023), AUC-IoU provides consistent characterization by eliminating threshold artifacts. It also extends beyond benchmarks such as Saliency-Bench and medical imaging evaluations Saporta et al. (2022); Zhang et al. (2023), which identified attribution method limitations but not the underlying evaluation bias. The size-stratified analysis further reveals dependencies that prior clinical studies may have obscured through single-threshold evaluation, e.g., XRAI's superiority across lesion sizes and GradCAM's 269% improvement from small to large lesions Cerekci et al. (2024); Wollek et al. (2023).

## 5.2 THEORETICAL UNDERSTANDING OF ATTRIBUTION BEHAVIOR

Systematic threshold response patterns provide insight into attribution mechanisms. Gradient-based methods show monotonic performance degradation with increasing thresholds, reflecting concentrated high-magnitude attributions that are penalized by aggressive binarization. This aligns with Integrated Gradients' theoretical basis of evidence accumulation along paths, which naturally yields focused feature importance. In contrast, LIME exhibits threshold-invariant performance, producing diffuse, uniform attribution distributions consistent with its local linear modeling on superpixels. These distinct profiles imply application-dependent suitability: concentrated methods for precise feature identification, diffuse methods for capturing broader feature relationships.

## 5.3 BROADER AND CLINICAL IMPLICATIONS

Threshold bias exemplifies broader evaluation challenges in machine learning where metrics embed hidden assumptions. This parallels issues such as confidence thresholding in classification or hyperparameter sensitivity in model comparisons. The threshold-free framework provides a template for mitigating such biases, ensuring robust conclusions across ML domains. Clinically, size-stratified analysis shows that aggregate performance metrics mask substantial variation (0–269% improvement factors). For small lesions, the most difficult diagnostic task, XRAI significantly outperforms GradCAM (AUC-IoU: 0.106 vs. 0.046), underscoring that method selection cannot rely on global averages. These results suggest adaptive explanation systems that dynamically select attribution methods based on case characteristics, rather than applying a single method universally.

## 5.4 LIMITATIONS AND RECOMMENDATIONS

This study is limited to a single dataset and binary classification task; generalization to other modalities, multi-class settings, and non-medical domains requires further validation. Threshold sensitivity may vary across contexts, demanding domain-specific analysis. The computational overhead of threshold-free evaluation ($19\times$ metric calculations) poses practical challenges for large-scale studies, motivating development of efficient approximations. Moreover, IoU alone may not fully capture attribution quality; future work should examine threshold bias in faithfulness, human evaluation, and downstream task metrics. For the XAI community, we recommend adopting threshold-free evaluation as standard practice, particularly in high-stakes settings. Method comparison studies should report threshold sensitivity analyses to expose bias effects. Benchmarks should incorporate threshold-free protocols, and domain-specific guidelines should address application-relevant evaluation needs. Finally, ensemble approaches that combine complementary strengths revealed by comprehensive evaluation may prove more reliable than reliance on single attribution techniques.

## 6 CONCLUSION

This work demonstrates that arbitrary threshold selection introduces systematic bias in attribution evaluation depending solely on threshold choice. Our threshold-free AUC-IoU framework eliminates this artifact, enabling reliable method comparison that reveals XRAI's consistent superiority across lesion sizes and statistically validated performance differences. The observed threshold-response patterns clarify fundamental attribution behaviors: gradient-based methods concentrate attributions optimal at low thresholds, while perturbation-based approaches remain threshold-invariant. Size-stratified analysis further shows that method selection cannot rely on aggregate metrics alone. Beyond XAI, this work exemplifies broader ML evaluation challenges where hidden assumptions bias results. We recommend adoption of threshold-free evaluation, development of domain-specific guidelines, and exploration of ensembles that utilize complementary strengths across attribution methods.

## 7 REPRODUCIBILITY STATEMENT

We ensured reproducibility by fixing random seeds (Python 3.11, PyTorch 2.7.1; seed=42) and using deterministic PyTorch settings. Dataset source, preprocessing steps, and splits are described in the Methodology and Appendix A. All model hyperparameters, training procedures, and evaluation

protocols (including the 19-threshold sweep and statistical tests) are specified in the main text and appendix. These details are sufficient for an independent researcher to replicate our results without additional resources.

## 8 ETHICS STATEMENT

We use the publicly available HAM10000 dataset solely for research. The images are de-identified and provided under research-friendly terms; we adhere to the dataset's stated license and usage conditions. Our work is methodological and does not make diagnostic claims. Saliency/attribution maps should not be used as a standalone basis for clinical decision-making. No additional patient data, personally identifiable information, or demographic attributes were collected or used.

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

## A  APPENDIX

## B  DATASET AND MODEL TRAINING DETAILS

### B.1  DATASET PREPROCESSING

- Source: HAM10000 dataset, 10,015 dermoscopic images with binary segmentation masks.
- Images resized to 224×224 using bilinear interpolation; pixel intensities normalized to [0,1] and standardized with ImageNet mean/std.
- Segmentation masks binarized at threshold 127.
- Stratified split: 70% train, 15% validation, 15% test, preserving melanoma prevalence ( 11%).
- Attribution evaluation subset: 500 test images (167 melanoma, 333 non-melanoma) to ensure statistical power and minority-class coverage.

### B.2  MODEL ARCHITECTURE AND TRAINING

- Base model: ResNet-18 pretrained on ImageNet.
- Architecture: final FC layer modified from 512 to 2 units; conv1–layer3 frozen, layer4 + classifier fine-tuned ($\sim$8.39M trainable parameters).
- Loss: class-weighted cross-entropy (non-melanoma 0.563, melanoma 4.499).
- Optimizer: Adam, learning rate $1 \times 10^{-4}$.
- Early stopping: patience=5 epochs, $\delta$=0.1% minimum validation improvement.
- Training converged after 15 epochs; best validation accuracy=92.61%.

### B.3  PROBABILITY CALIBRATION

- Applied temperature scaling using validation logits.
- Optimal temperature $T^* = 2.28$ (via L-BFGS).
- Reduced NLL from 0.292 to 0.208; mean maximum probability from 96.9% to 91.8%.
- Calibration particularly important for LIME, which relies on probability estimates for perturbation-based explanations.

## C  ADDITIONAL RESULTS FIGURES

We include supplementary visualizations supporting the main results.

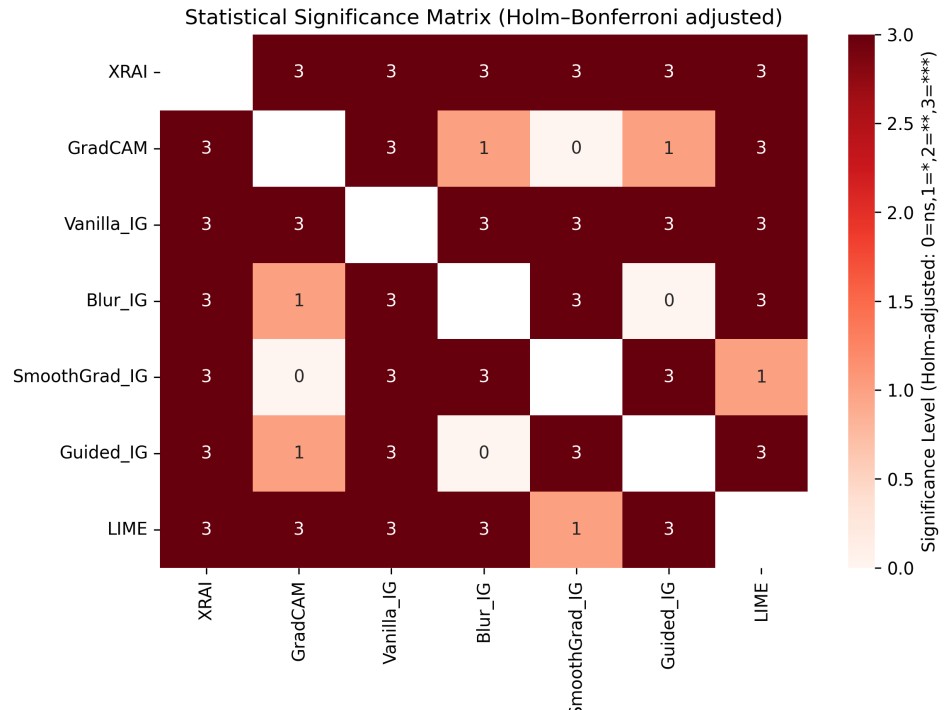

Figure 2: Statistical significance matrix for pairwise method comparisons after Holm-Bonferroni correction (n=500 images). Color coding: 0=non-significant (white), 1=$p < 0.05$ (light red), 2=$p < 0.01$ (medium red), 3=$p < 0.001$ (dark red). XRAI shows consistent superiority over all other methods (entire top row in dark red), while several method pairs show no significant differences (GradCAM vs. SmoothGrad IG, Blur IG vs. Guided IG), indicating that apparent performance rankings can be misleading without proper statistical validation.

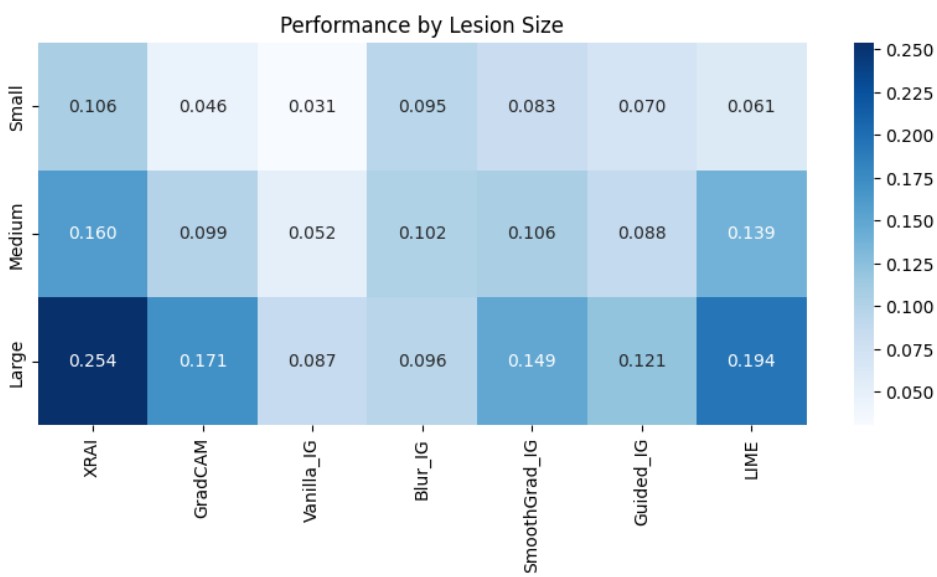

Figure 3: Method performance across small ($\leq$33rd percentile, n=133), medium (33rd–67th percentile, n=160), and large ($\geq$67th percentile, n=207) lesions using AUC-IoU scores. XRAI maintains consistent superiority across all size categories, while GradCAM shows size sensitivity (269% improvement from small to large lesions). Blur IG exhibits size-invariant performance, demonstrating fundamental differences in how attribution mechanisms respond to lesion scale characteristics.

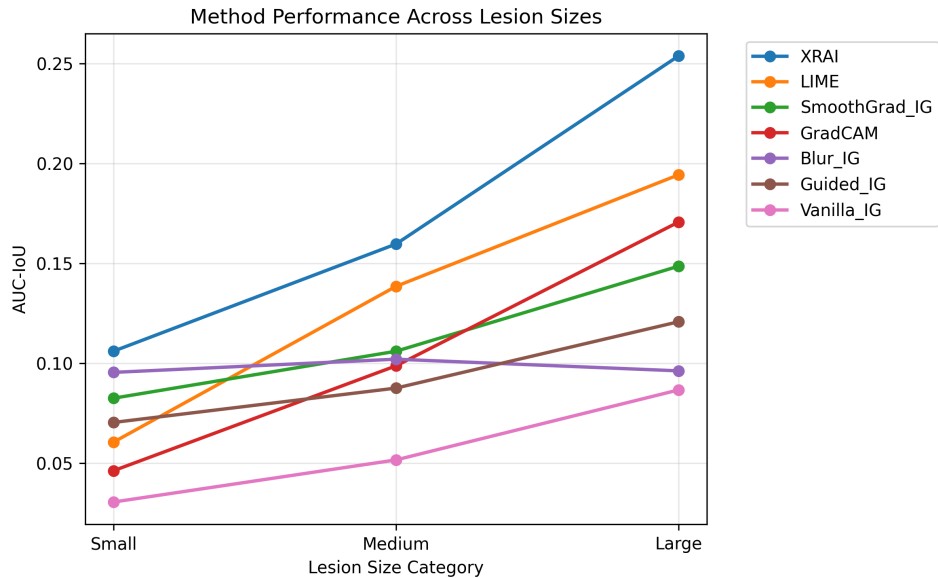

Figure 4: Linear trend analysis revealing distinct attribution profiles across lesion sizes. Steep upward slopes for XRAI and GradCAM contrast with Blur IG's flat trajectory, indicating that gradient-based and region-based methods scale better with lesion size compared to noise-reduction approaches. These distinct scaling behaviors have direct implications for clinical deployment, particularly for challenging small lesion detection scenarios.

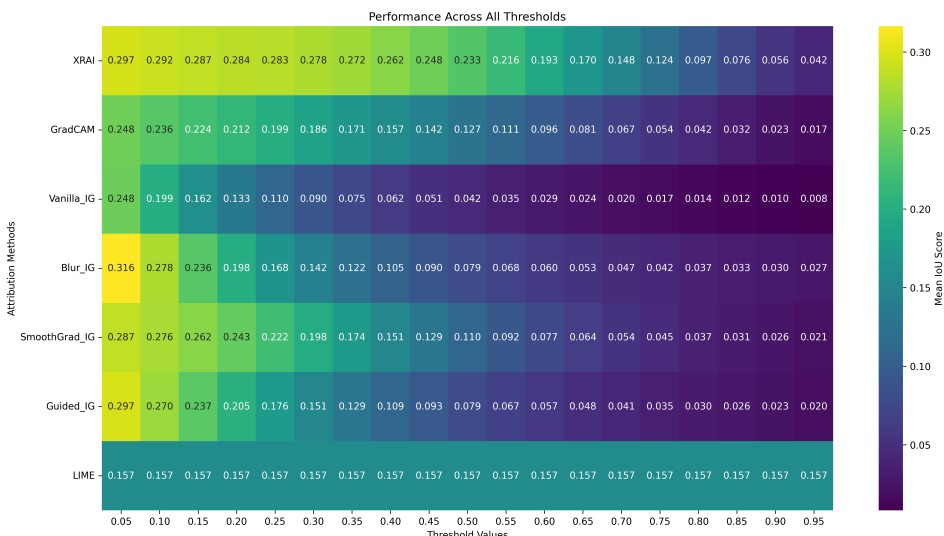

Figure 5: Complete threshold spectrum analysis showing method performance across 19 uniformly spaced thresholds ($\tau \in [0.05, 0.95]$) using color-coded IoU scores. Gradient-based methods exhibit monotonic performance degradation with increasing thresholds (blue to yellow transition), while LIME demonstrates threshold-invariant behavior (consistent green). This visualization demonstrates how arbitrary threshold selection can completely reverse method rankings, with performance swings exceeding 200 percentage points for individual methods.

# D ATTRIBUTION METHOD VISUALIZATIONS

Representative examples of attribution methods demonstrating the distinct response patterns that contribute to threshold sensitivity in our evaluation framework.

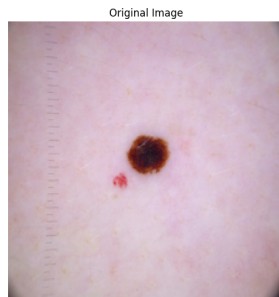
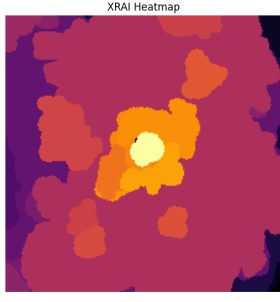
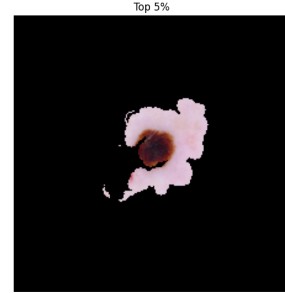

Figure 6: XRAI attribution example on dermatological image. Left: Original image with dark lesion. Center: XRAI heatmap with yellow indicating high attribution weight. Right: Top 5% threshold binarization. XRAI produces coherent region-based attributions aligned with lesion boundaries.

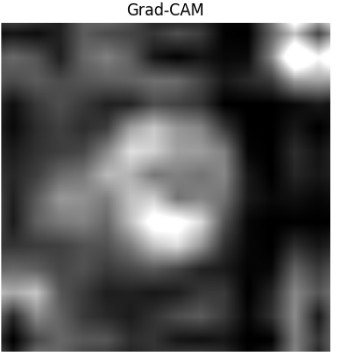
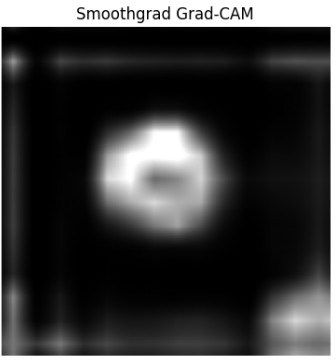

Figure 7: Activation-based method comparison. Left: Standard Grad-CAM showing broad activation patterns. Right: SmoothGrad Grad-CAM with noise reduction producing more focused attributions through averaging across noisy input versions.

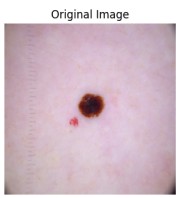
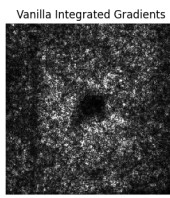
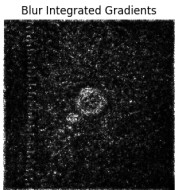
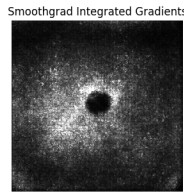
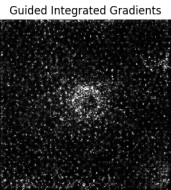

Figure 8: Integrated Gradients variants comparison. From left: Original image, Vanilla_IG, Blur_IG, SmoothGrad_IG, Guided_IG. Each variant exhibits distinct attribution characteristics: Vanilla_IG shows noisy concentrated patterns, Blur_IG produces focused circular responses, SmoothGrad_IG generates smoother distributions, and Guided_IG creates sparse high-contrast features. These distinct patterns explain the threshold-dependent performance variations observed in our evaluation.

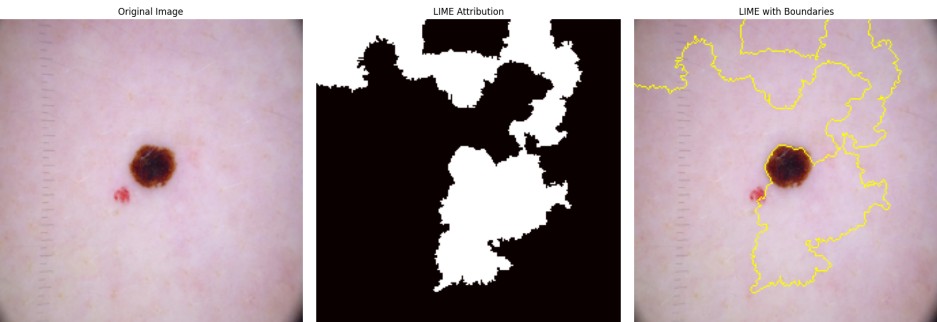

Figure 9: LIME attribution example demonstrating threshold-invariant behavior. Left: Original image. Center: LIME superpixel-based attribution map. Right: LIME with segment boundaries highlighted. Unlike gradient-based methods, LIME's discrete superpixel approach produces threshold-invariant performance, explaining its consistent ranking across different evaluation thresholds.

