# OpenReview forum: "Systematic Evaluation of Attribution Methods: Eliminating Threshold Bias and Revealing Method-Dependent Performance Patterns"
_ICLR.cc/2026/Conference — ICLR 2026 Conference Withdrawn Submission_

### Official Review · Reviewer_iXrY · 2025-10-16

**Soundness:** 2
**Presentation:** 1
**Contribution:** 2
**Rating:** 0
**Confidence:** 5

**Summary:**

This paper is incomplete and not ready for review. The paper does not clearly explain what threashold bias problem it is solving. Moreover, it appears that entire subsections are missing,  3.4 ATTRIBUTION EVALUATION FRAMEWORK and 3.4.1 THRESHOLD-FREE EVALUATION PROTOCOL have no content.

**Strengths:**

The assessment of attribution methods is a challenging problem.

**Weaknesses:**

The paper is incomplete
The main threshold challenge is not clearly described

**Questions:**

n/a

---

### Official Review · Reviewer_UP6x · 2025-10-27

**Soundness:** 1
**Presentation:** 1
**Contribution:** 2
**Rating:** 0
**Confidence:** 5

**Summary:**

This paper proposes a new ground-truth attribution evaluation metric which aims to solve a longstanding issue for measuring GT IoU. In these existing metrics, attribution maps are binarized with a threshold for their evaluation, but the choice of threshold can lead to a biased score, and no one threshold is proper for all attributions. To fix this, they propose to evaluate each attribution over many thresholds to create an AUC IoU metric.

**Strengths:**

Ground-truth evaluation is a widely used strategy and is well known to suffer from the threshold bias. This paper makes an interesting contribution by aiming to solve this problem.

What is available of the seemingly incomplete methodology indicates a decent approach to solving the threshold selection problem.

There is quantitative proof that this is a more reasonable approach than single threshold IoU.

**Weaknesses:**

The paper is clearly unfinished. The methodology appears to have a critical section (4.1) empty, making the entire definition of the method only 17 lines.

Tables are poorly formatted and out of bounds. Many of them are also confusing to interpret. The current in-text citations should be expanded to give the reader more intuition for how to interpret the results.

Only one model and one dataset was used (ResNet-18 and HAM10000). ResNet-18 is very small. I would want to see this evaluation on larger CNNs and perhaps even a fine-tuned ViT model for HAM10000. Evaluation on the ImageNet segmentation GT dataset [1] following [2,3] should also be included for more variation.

The selection of attribution methods could be expanded. What is included is fine given the other items that need more improvement, but the Captum library is accessible and easy to use and its use will allow a significant improvement in experimental scope.

There is quantitative proof that this method is better than single-threshold IoU metrics, but the proof is lacking. Showing the variations in score under different thresholds is helpful, but I recommend considering using the metric evaluations from [4] in addition to what is present.

[1] ImageNet auto-annotation with segmentation propagation

[2] Transformer interpretability beyond attention visualization

--> [3] https://github.com/hila-chefer/Transformer-Explainability (/data, /utils, and /baselines/ViT/imagenet_seg_eval.py)

[4] Sanity checks for saliency metrics

**Questions:**

LIME performs perturbation but is not a true perturbation-based method. Its binary output leads to the static behavior under changing thresholds, which is interesting, but it would be great to see how a better representative perturbation method (any method based on feature ablation, occlusion, or SHAP) performs.

I would want to see what the rankings of the attributions are when you pick the "ideal" threshold for each. Since you are already evaluating over 19 thresholds, I am suggesting that you report for each method, on each image, the maximum IoU score over the 19 thresholds. This will help contextualize the average performance over all the thresholds.

Is 4.1 unfinished? Why is it there? What is missing from this section?

Overall, I could not accept this paper in its current state, it should probably be a desk reject from a formatting perspective. It is clearly unfinished and was not given proper care before submission. The idea is interesting, but I believe the flaws cannot be corrected in a rebuttal period. I strongly recommend the authors take more time to build the mathematical foundation and evaluation of their method and resubmit the paper in a finished form such that its full potential can be realized.

---

### Official Review · Reviewer_QBfF · 2025-10-28

**Soundness:** 1
**Presentation:** 2
**Contribution:** 1
**Rating:** 2
**Confidence:** 5

**Summary:**

The authors identify that the specific threshold used during attribution evaluation has a significant impact on the outcome of the evaluation, which is not desirable because the threshold is outside noise influencing attribution performance on the metric used. The authors propose the Area Under the Curve for Intersection over Union (AUC-IoU) metric, which does not use any thresholds when ranking attributions. The authors provide an empty methodology section, so I am unable to provide an overview of the methodology. Some results measuring different attributions on the proposed metric are given, as well as some statistical significance analysis.

**Strengths:**

- Originality: The idea seems novel, but I am unable to asses it because the section on the metric is blank.
- Quality / Clarity: The introduction and motivation are clear. The presented tables and figures are quality.
- Significance: The metric seems significant, but I am unable to asses it because the section on the metric is blank.

Other Notes:
- The motivation for the proposed method is backed up by previous work.

**Weaknesses:**

- The section on the metric is blank, so I cannot adequately asses the metric.
- The experiments evaluate different attribution methods on the given metric, which is fine. However, there are no experiments that compare against other metrics or give any insight into why the proposed metric is better. Table 5 claims to compare against other metrics, but these are very simple metrics that I do not believe are often used.
- The dataset used in evaluation seems to only have 2 classes. This is not a significant enough evaluation. More datasets with more classes should be evaluated over.
- The same sample of melanoma positive cases is used through the validation, testing, and attribution evaluation splits. This ignores the reasoning behind different data splits.

**Questions:**

- I would suggest the authors look at other evaluation metrics and compare against those. For example, the insertion/deletion tests in [1] and [2].
- I would also suggest the authors look into other datasets like Imagenet [3] or the German Traffic Sign Dataset [4].

[1] Vitali Petsiuk, Abir Das, and Kate Saenko. Rise: Randomized input sampling for explanation of black-box models. In Proceedings of the British Machine Vision Conference (BMVC), 2018.

[2] A. Kapishnikov, T. Bolukbasi, F. Viegas, and M. Terry. Xrai: Better attributions through regions. In 2019 IEEE/CVF International Conference on Computer Vision (ICCV), pages 4947–4956, Los Alamitos, CA, USA, nov 2019. IEEE Computer Society.

[3] J. Deng, W. Dong, R. Socher, L. -J. Li, Kai Li and Li Fei-Fei, "ImageNet: A large-scale hierarchical image database," 2009 IEEE Conference on Computer Vision and Pattern Recognition, Miami, FL, USA, 2009, pp. 248-255, doi: 10.1109/CVPR.2009.5206848.

[4] J. Stallkamp, M. Schlipsing, J. Salmen, and C. Igel. The German Traffic Sign Recognition Benchmark: A multi-class classification competition. In Proceedings of the IEEE International Joint Conference on Neural Networks, pages 1453–1460. 2011.

Final Review: The paper is missing the section describing the proposed metric, making this paper not suitable for acceptance. Beyond that, the experimental evaluation does not demonstrate strong performance of the proposed metric against other metrics or provide any insight into why it would perform well. For these reasons, I feel the need to reject the paper (2/10).

---

### Official Review · Reviewer_HzL2 · 2025-10-31

**Soundness:** 1
**Presentation:** 2
**Contribution:** 1
**Rating:** 2
**Confidence:** 4

**Summary:**

In XAI, the attribution method explains the prediction of a neural network by highlighting the influential input features, which are then binarized by a certain threshold. In this way, the evaluation might be unfair because of the threshold selection bias. In this work, the authors proposed a threshold-free framework that computes AUROC and IoU across the full threshold spectrum. They showcased that the single threshold metrics might lead to contradictory results, but their proposed method achieved a reliable differentiation. They tested their proposed method on XRAI, LIME and various integrated gradient methods.

**Strengths:**

1. The evaluation of XAI methods is an interesting topic that definitely needs a bit more attention. I appreciate the effort the authors put into this direction.
2. I believe that the reliance on a threshold for the evaluation of XAI is indeed an issue for fairer comparison. The research question is solid and worth putting effort into. I would like to encourage the authors to further dig into this question, and I hope my feedback could be helpful.

**Weaknesses:**

1. I am not super convinced by their conclusion that the proposed threshold-free method is better based on their experiments. In this work, they are using the mask as the ground truth (GT) explanation, but why? To my understanding, it is also okay if the NN is not looking at the whole skin lesion area but just a part of it, or even that the features of the specific disease are only a part of the segmentation. Also, they used the relative difference as a kind of measurement for the soundness of the explanation. However, it only measures how the proposed method differs from the IoU under a certain threshold; I am not sure how it indicates a better evaluation.

2. Limited novelty/contribution. Though I do believe this topic is timely and interesting, this work seems to be very limited in contribution. Apart from the abovementioned limitation about the validation of the proposed method, the authors only tested their method on one dataset with one model only. The model being tested is also quite a basic model such as ResNet-18. A simple method itself is not a problem; however, with an unclear validation and this amount of experiments, I found the contribution quite limited.

3. Lack of literature in evaluation of explainability methods. There is literature about XAI evaluation. Since this work is more about proposing a new method for evaluating explanations, I encourage the authors to include that work and establish their validation based on it. For example, [1] and [2].

4. Presentation of the work. I believe the authors could do a better job at presenting this work, e.g., better table formatting, better illustrations, clearer description of the methods, and more discussion.

5. Seems unfinished in Section 3.4.

[1] Nauta, M., Trienes, J., Pathak, S., Nguyen, E., Peters, M., Schmitt, Y., ... & Seifert, C. (2023). From anecdotal evidence to quantitative evaluation methods: A systematic review on evaluating explainable ai. ACM Computing Surveys, 55(13s), 1-42.
[2] Kadir, M. A., Mosavi, A., & Sonntag, D. (2023, July). Evaluation metrics for xai: A review, taxonomy, and practical applications. In 2023 IEEE 27th International Conference on Intelligent Engineering Systems (INES) (pp. 000111-000124). IEEE.

**Questions:**

1. Why did you choose the segmentation mask as the GT for explanations, and why does that choice make sense?
2. How does the relative difference demonstrate that your proposed method is more reliable?
3. Regarding the selection bias in threshold choice, would it be possible to illustrate this issue using your skin lesion dataset?

---

### Note · Authors · 2025-11-27

I have read and agree with the venue's withdrawal policy on behalf of myself and my co-authors.